

# EPR and SQUID interrogations of Cr(III) trimer complexes in the MIL-101(Cr) and bimetallic MIL-100(Al/Cr) MOFs

Kavipriya Thangavel[1,2], Andrea Folli[2], Michael Ziese[1], Steffen Hausdorf[3], Stefan Kaskel[3], Damien M. Murphy[2], and Andreas Pöppl[1*]

**1** Felix Bloch Institute for Solid State Physics, University of Leipzig, Linnestrasse. 5, 04103 Leipzig, Germany
**2** School of Chemistry, Cardiff University, Park Place, Cardiff CF10 3AT, UK
**3** Department of Inorganic Chemistry, Technische Universität Dresden, Bergstrasse 66,01069 Dresden, Germany

★ poeppl@physik.uni-leipzig.de

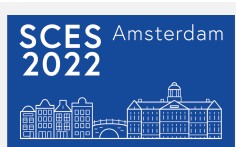

## Abstract

Herein, electron paramagnetic resonance (EPR) spectroscopy at X- (9.4 GHz), Q- (34 GHz) and W-band (95 GHz), and superconducting quantum interference device (SQUID) measurements on antiferromagnetically coupled metal trimers in MIL-101(Cr) and MIL-100($Al_{0.8}Cr_{0.2}$) MOFs were investigated. At low temperatures, the Cr(III) trimers exhibit a Dzyaloshinsky-Moriya (D-M) interaction, and have a total spin state $S_T = 1/2$.

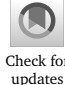
# 1  Introduction

Metal-organic frameworks (MOFs) are a novel and the most prominent class of microporous materials for the applications such as gas storage and separation, catalysis, heat storage, sensing, electrochemical energy, supercapacitors and liquid purification, owing to their unique structural diversity and tunability [1–5]. The structured and tunable pore nature, enormous surface and volume area, crystal integrity and low density are just some of the salient properties of MOFs, which make MOFs revolutionary materials for numerous applications including device fabrication, chemical and technological industries [3–5]. In recent years, the magnetic behaviours of MOFs have been getting attention in the field of molecular magnetism since they consist of a paramagnetic secondary building unit (SBU) along with the organic connecting linker [6]. Among MOFs, MIL (Materials Institute Lavoisier) -101 and MIL-100 MOF families are complex MOFs in crystal structure just with a single linker and SBU. In addition, these MOFs have pores in two different sizes [7] (29 Å and 34 Å  for MIL-101; 25 Å  and 29 Å  for MIL-100) that allow the adsorption of gas and liquid [8,9](Figure 1a). MIL-101(M) and MIL-100(M) frameworks contain trinuclear metal ions $M(III)_3$ in the octahedral units, and four trimers, forming a supertetrahedron. In MIL-100, the trimers are connected through benzene tricarboxylate (BTC) linker and in MIL-101, the linker is benzene dicarboxylate (BDC) [8]. Understanding the pairwise magnetic exchange interaction in the metal trimer clusters is complicated, and the term magnetic 'frustration' (Figure 1b) often describes this situation [10]. The influence of anti-symmetric exchange on the electronic properties of trinuclear $Cu(II)_3$ metal complexes was experimentally observed for the first time by Tsukerlat *et al.* [10, 11]. Likewise, the antisymmetric exchange interaction of $Cr(III)_3$ trimer cluster was experimentally investigated by M. Honda *et al.*., [12] A. Vlachos *et al.*., [13] A. Figuerola *et al.*., [14] by means of superconducting quantum interference device (SQUID) and electron paramagnetic resonance (EPR) spectroscopy. Furthermore, the role played by the guest-framework intermolecular interactions, the influence of adsorption on the intramolecular interactions and the changes in the internal structure can be elucidated by tracking the paramagnetic species in the SBU [6,9,15–18]. In this regard, EPR spectroscopy is one of the inevitable tools to understand the change in the local structure of MOFs due to guest molecular interaction during *ex situ* [16] and *in situ* [15,17,19] gas adsorption, liquid adsorption, light irradiation [20], post synthetically modified ion exchange, [2] and intermolecular magnetic couplings [2,21–23] by monitoring the behavior of paramagnetic metal ions in the SBU.

Herein, we studied and discussed the local structure and the intramolecular interaction of EPR active Cr(III) trimer complex of the MIL-101(Cr) and magnetically diluted bimetallic MIL-100($Al_{0.8}Cr_{0.2}$) MOFs with $Al(III)_{3-x}Cr(III)_x$ units for the comparison by means of SQUID magnetometer and multi-frequency EPR spectroscopy techniques.

# 2  Experimental Details

Metal-organic frameworks MIL-101(Cr) and MIL-100($Al_{0.8}Cr_{0.2}$) were purchased from commercial MOF seller 'Materials Center, Technische Universität Dresden'. In MIL-100(Al/Cr), 20% Cr has been incorporated on Al sites within the frameworks. All the experiments mentioned below were done on the as-synthesized MIL-101(Cr) and MIL-100($Al_{0.8}Cr_{0.2}$) MOFs.

The magnetization measurements were performed using a SQUID magnetometer (Quantum Design MPMS XL). Hysteresis loops at several temperatures between ± 7 T as well as the temperature-dependence at 0.5 T in the temperature range of 4 K to 300 K, were measured.

Continuous wave (cw) X-band (∼ 9.5 GHz) EPR spectra were measured at temperature ranging from $T = 7$ K to $T = 290$ K by means of a Bruker EMXmicro spectrometer fitted with a

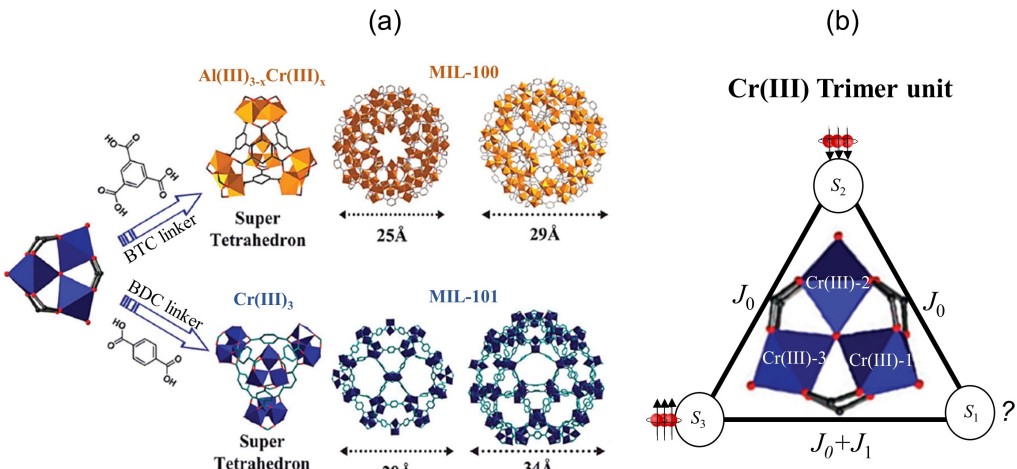

Figure 1: (a) Cr(III)/Al(III) ions are in the octahedral trimer units and four trimers, forming a supertetrahedra (Modified from rivera *et al.* [24] with permission from the Royal Society of Chemistry.) showing MIL-101(Cr) and MIL-100(Al$_{0.8}$Cr$_{0.2}$) frameworks, and (b) Scheme of the trimer unit contains three $S = 3/2$ Cr(III) spins shows the situation of spin 'frustration' ($J_0$ and $J_1$ are the exchange coupling parameters).

Bruker ER4119HS cylindrical cavity using an He cryostat ESR900, Oxford instruments. In all X-band experiments, the microwave (mW) power was set to 2 mW, modulation frequency was kept as 100 kHz, and modulation amplitude was maintained to 10 G to acquire a spectrum without any line shape distortion and saturation. Cw Q-band (∼34 GHz) EPR spectra were performed using Bruker EMX 10-40 spectrometer fitted with cylindrical cavity and an Oxford Instruments CF935 cryostat at $T = 20$ K and 295 K. In Q-band, the experimental parameters, the mW power, modulation frequency and modulation amplitude were set to 1.8 mW, 100 kHz and 20 G, respectively. Unlike X- and Q-band EPR spectrometers, the high magnetic field of W- band (∼95 GHz) EPR requires a superconducting magnet (Bruker 6T SC) and measured using an Elexsys E600 spectrometer equipped with a Bruker E600-1021H TeraFlex resonator, and the spectra were recorded at $T = 20$ K and 300 K. For W-band experiments, the microwave power kept as either 5 $\mu$W or 50 $\mu$W depending on the signal quality. The EPR intensities of the X-band signals ranging from $T = 7$ K to $T = 280$ K (Figure 3) were extracted by taking double integration of the full EPR spectrum.

In general, the exchange Hamiltonian for the Cr(III) trimers can be written as [12, 25],

$$\hat{\mathscr{H}}_{Ex} = -2J_0 \left( \hat{\vec{S}}_1 . \hat{\vec{S}}_2 + \hat{\vec{S}}_2 . \hat{\vec{S}}_3 + \hat{\vec{S}}_3 . \hat{\vec{S}}_1 \right) - 2J_1 \left( \hat{\vec{S}}_3 . \hat{\vec{S}}_1 \right), \tag{1}$$

where $\hat{\vec{S}}_1$, $\hat{\vec{S}}_2$ and $\hat{\vec{S}}_3$ are the Cr(III) spin operators with single $S_i = 3/2$ spins for each chromium ion. $J_0$ is the main exchange parameter and $|J_1/J_0| \ll 1$. Eqn.1 results in five degenerate energy levels corresponding to the total spin states $S_T = 1/2, 3/2, 5/2, 7/2$ and $9/2$ of the trimer with the twofold degenerate $S_T = 1/2$ state being the ground state in the case of antiferromagnetically (AFM) coupled trimers when $J_1 = 0$. The degeneracy of the $S_T = 1/2$ state will be further lifted only when $J_1 \neq 0$ [12]. However, $J_1$ is considered to be negligible since Cr(III) ions in the trimer unit is assumed to be in equal distance.

Also, the Hamiltonian of inter-trimer interaction can be defined by [12, 25],

$$\hat{\mathscr{H}}_{in} = -2J_{in} \left( \hat{\vec{S}}_{iT} . \hat{\vec{S}}_{jT} \right), \tag{2}$$

where $S_{iT}$ and $S_{jT}$ correspond to the total spins of $i$-th and $j$-th trimers, respectively. In case

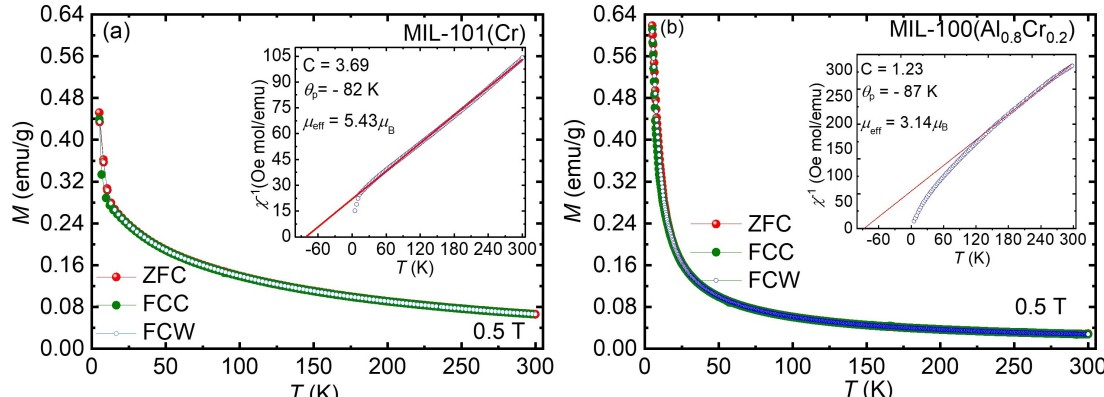

Figure 2: ZFC, FCC and FCW *M-T* curves of (a) MIL-101(Cr), and (b) MIL-100($Al_{0.8}Cr_{0.2}$) with an applied field of 0.5 T. (Insets: The temperature-dependence of the reciprocal of the magnetic susceptibility of its corresponding ZFC curves.)

of EPR experiments, the additional terms

$$\hat{\mathscr{H}} = \mu_B \vec{B} \boldsymbol{g}(\hat{\vec{S}}_1 + \hat{\vec{S}}_2 + \hat{\vec{S}}_3) + \sum_{i=1}^{3}\left(D_{ZFS,i}\left\{\hat{S}^2_{i,z'} - S_i[S_i+1]/3\right\} + E_{ZFS,i}\left\{\hat{S}^2_{i,x'} - \hat{S}^2_{i,y'}\right\}\right) +$$
$$D\left(\hat{\vec{S}}_1 \times \hat{\vec{S}}_2 + \hat{\vec{S}}_1 \times \hat{\vec{S}}_3 + \hat{\vec{S}}_2 \times \hat{\vec{S}}_3\right), \tag{3}$$

must be considered in the spin-Hamiltonian for the Cr(III) trimers, where the first term represents the Zeeman interaction between $S = 3/2$ electron spins of the chromium ions and the external magnetic field ($\mu_B$ is the Bohr magneton, $\boldsymbol{g}$ is the *g*-tensor, $\vec{B}$ is the external magnetic field). The second term corresponds to the zero field splitting (ZFS) with the axial and rhombic ZFS parameters $D_{ZFS,i}$ and $E_{ZFS,i}$ and the third term describes the antisymmetric exchange contribution caused by the Dzyaloshinsky-Moriya (D-M) interaction. Here $D$ is a pseudo-vector directed perpendicular to a plane defined by the Cr(III) ions in the trimers [12].

## 3 Results and Discussion

### 3.1 SQUID magnetometry results

Figure 2 shows the temperature variation of zero-field-cooling (ZFC), field cooled cooling (FCC) and field cooled warming (FCW) magnetic susceptibility behavior of MIL-101(Cr) and MIL-100($Al_{0.8}Cr_{0.2}$) MOFs obtained in 0.5 T magnetic field. The *M-T* (magnetization vs temperature) behavior of both the MOFs suggests a paramagnetic nature. The paramagnetic susceptibility of both the MOFs follows Curie-Weiss (C-W) law (eqn. 4) for temperatures $T > 15$ K (MIL-101(Cr) and $T > 120$ K (MIL-100($Al_{0.8}Cr_{0.2}$)). The fitted parameters (Figure 2a & 2b insets) yield effective paramagnetic moment ($\mu_{eff}$) and paramagnetic Curie temperature ($\theta_p$) values as -82 K and 5.43 $\mu_B$/f.u. (f.u.-formula unit), respectively for MIL-101(Cr) MOF. While for MIL-100($Al_{0.8}Cr_{0.2}$), it is -87 K and 3.14 $\mu_B$/f.u., respectively. The occurrence of a negative sign of $\theta_p$ of both the MOFs indicates the existence of AFM interactions in the system.

The magnetic susceptibility can be written as

$$\chi = \frac{C}{T - \theta_p}, \tag{4}$$

where, $\chi$ is the magnetic susceptibility, $C$ is the Curie constant, and $\theta_p$ is the paramagnetic Curie temperature.

Also, $\mu_{eff}$ was calculated using the following relation,

$$\mu_{eff} = \sqrt{\frac{3k_B C}{N\mu_B^2}} = \sqrt{8C}\,, \tag{5}$$

where, $\mu_{eff}$ is the effective Bohr magneton, $k_B$ is the Boltzmann constant, $C$ is the Curie constant, $N$ is the Avogadro number and $\mu_B$ is the electron Bohr magneton. ZFC, FCC, and FCW curves for both materials are identical (Figure 2), and No magnetic hysteresis effects at any temperatures were observed, as shown by field-dependent magnetization measurements (Figure 5 in Appendix A).

For MIL-101(Cr) the overall temperature-dependence of $\chi^{-1}$ is typical for Cr(III) trimers [12, 13, 26]. Therefore, using [12, 13, 26]

$$\theta_p = 5\frac{\left(J_0 + \frac{J_1}{3}\right)}{k_B}\,, \tag{6}$$

we can estimate $J_0$ = -11.4 cm$^{-1}$ for MIL-101(Cr) from the susceptibility data at T > 15 K. Here, we assumed $J_0 \gg J_1$ as we could not determine $J_1$ from the magnetization measurements. According to Honda *et al.*, [12] the steep drop in $\chi^{-1}$ at low temperatures T < 15 K can be associated with the inter-trimer interaction and provides a lower limit for a second Curie temperature-like parameter $\theta'_p$ > -6 K (Figure 6 in Appendix A) where

$$\theta'_p = \frac{nJ_{in}}{4k_B}\,. \tag{7}$$

In the case of MIL-101(Cr) with its supertetrahedral structure of the connected Cr(III) trimers [8] n = 3 provides $|J_0| > |J_{in}|$ ($J_{in}$ = -5.6 cm$^{-1}$).

In the case of MIL-100(Al$_{0.8}$Cr$_{0.2}$), the $\chi^{-1}$ temperature-dependence does not give an indication of the presence of Cr(III) trimers but displays a rather smooth gradual drop (see Figure 6) response of one or several AFM coupled spin systems. Assuming again a further contribution dominating the magnetization data at T < 30 K and applying eqn. 4, we can roughly estimate the lower limit for a second Curie temperature like parameter $\theta'_p$ > -1.8 K (Figure 6 in Appendix A). The small limit $|\theta'_p|$ might indicate a very weak inter-trimer exchange between various Al(III)$_{3-x}$Cr(III)$_x$ units or just a simple paramagnetic contribution due to isolated Cr(III) ions.

## 3.2 EPR spectroscopy results

For comparison with magnetization data, the temperature-dependent EPR intensities $I_{EPR}$, which are proportional to the magnetic susceptibilities ($\chi \propto I_{EPR}$) were extracted from the temperature-dependent X-band EPR spectra (Figure 7 in Appendix B - which is derived by full double integration of the corresponding spectra). The insets of Figure 3 show their corresponding inverse intensity $I_{EPR}^{-1}$ as a function of temperature and fitted using C-W law, eqn. 4. The EPR intensities provide $\theta_p$ = -70 K and -85 K for MIL-101(Cr) and MIL-100(Al$_{0.8}$Cr$_{0.2}$), respectively, from the C-W fit which is in reasonable agreement with the $\theta_p$ values found from the magnetization susceptibility data and once again suggest the existence of antiferromagnetic interactions in both materials.

Figure 4 shows the multi-frequency EPR spectra of MIL-101(Cr) and MIL-100(Al$_{0.8}$Cr$_{0.2}$) MOFs at room and low temperatures. A full temperature dependence EPR X-band spectra for both samples are illustrated in Figure. 7 (Appendix B).

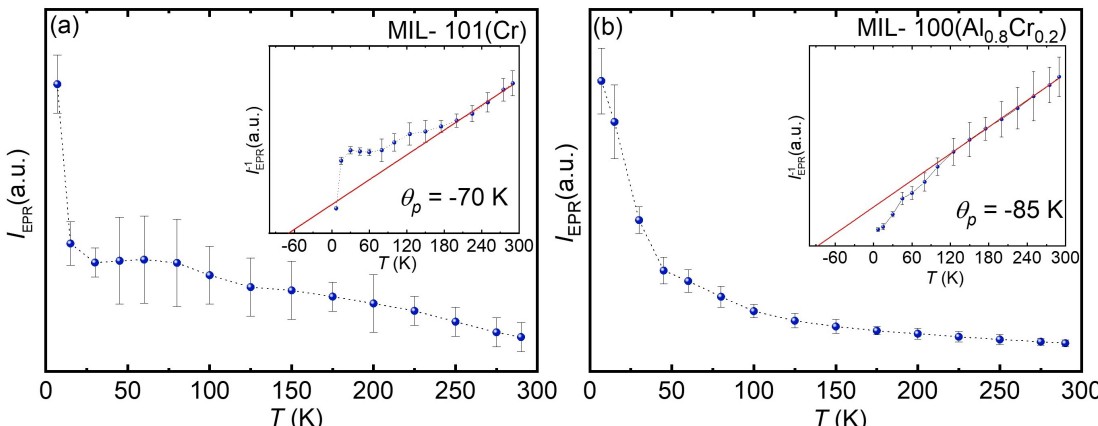

Figure 3: EPR Intensities ($I_{EPR}$) belong to Cr(III) spectra of (a) MIL-101(Cr), and (b) MIL-100($Al_{0.8}Cr_{0.2}$) extracted from temperature-dependent X-band EPR data. (Inset: Corresponding C-W fit of $I_{EPR}^{-1}$ as a function of temperature.)

Unlike MIL-100($Al_{0.8}Cr_{0.2}$), a significant shift in the $g$-value of the major signal A with $g_A = 1.979$ at 290 K towards smaller $g$-values of $g_E$, = 1.945 (signal E at 20 K) and $g_B = 1.779$ (signal B at 7 K) is observed for MIL-101(Cr) (Figure 4). The change in $g$ as a function of temperature for MIL-101(Cr) is given in Figure 8a (Appendix B) together with that of MIL- 100($Al_{0.8}Cr_{0.2}$) for comparison. We have to note that typical $g$-values for both isolated and coupled Cr(III) ions are in the range of $g = 1.97 - 1.98$ [27]. Honda *et al.* [12] discussed a comparable anomalous $g$-shift for Cr(III)$_3$ trimers in Cr-acetate and Cr-propionate. Such signals with characteristic $g$-shifts at low temperatures have also been reported for AFM coupled Cr(III)$_3$ trimers in other matrices [13, 26] and were assigned to their ground state with the total spin $S_T = 1/2$ [12, 13, 26].

The effective $g$-value of this doublet ground state is approximated by [12, 13, 26]

$$g_{eff} \approx g_A \sqrt{1 - \frac{48d^2}{(16J_1^2 + 48d^2) - (\mu_B \vec{B} g_A)^2}} . \tag{8}$$

Here, $d = D_{12,z} + D_{23,z} + D_{13,z}$ is the asymmetric exchange parameter of the trimer according to the D-M interaction and $\sqrt{16J_1^2 + 48d^2}$ corresponds to the splitting between the $S_T = 1/2$ ground state and the first excited state of the trimer having likewise $S_T = 1/2$. As $J_1$ is not known for MIL-101(Cr), we cannot derive the value of $d$ from our experimental results. However, the observed characteristic $g$-shift of the doublet ground state at $T < 30$ K can be considered as a signature for AFM coupled Cr(III)$_3$ trimers. On the other hand, the almost isotropic EPR signal at higher temperatures is less indicative of AFM coupling as all total spin levels $S_T = 1/2, 3/2, \ldots, 9/2$ will be more populated depending on the temperature and degeneracy.

The X-band EPR spectrum of MIL- 100($Al_{0.8}Cr_{0.2}$) (Figure 4a) appears to be more complex as at least two signals C and D having $g_C = 4.0 - 5.5$ and , $g_D = 1.973$, respectively can be discerned in the whole temperature range (Figure 7b in Appendix B). However, besides some decrease in the line-widths towards lower temperatures, (Figure 8b in Appendix B) the spectra reveal no characteristic temperature-dependence of the $g$ values of signals C and, in particular of signal D even at $T = 7$ K (Figure 8a in Appendix B). In addition, the line-width of signal D is significantly smaller than that of signal A in MIL-101(Cr) (Figure 8b). Therefore, we can exclude the formation of AFM coupled Cr(III)$_3$ trimers for MIL- 100($Al_{0.8}Cr_{0.2}$).

EPR spectra comparable to those of MIL- 100($Al_{0.8}Cr_{0.2}$) have been reported for Cr(III) doped phosphate glasses [27] and microporous materials such as silica and various zeolite

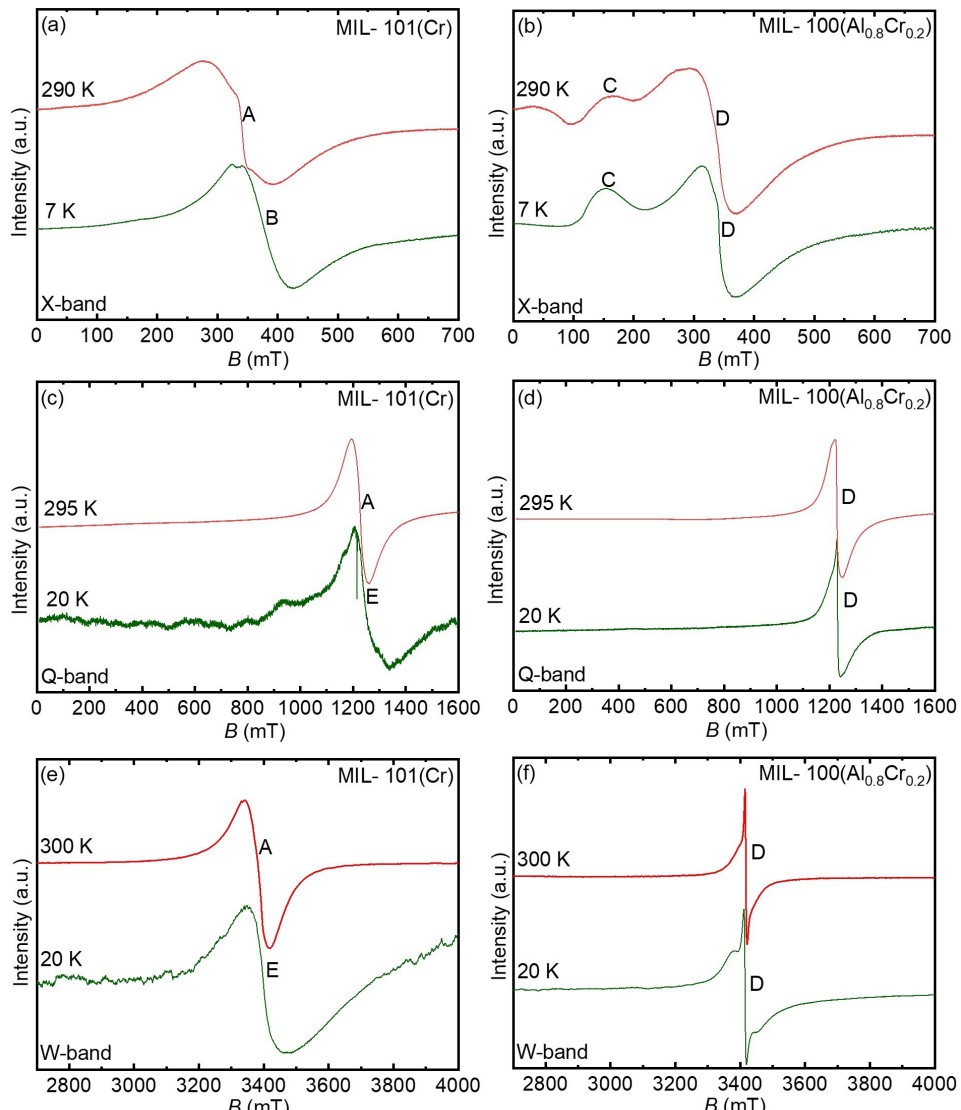

Figure 4: (a & b) X-band, (c & d) Q-band and (e & f) W-band EPR spectra of MIL-101(Cr) and MIL-100($Al_{0.8}Cr_{0.2}$) MOFs, respectively. ($g$-values of the corresponding signals: $g_A = 1.979$, $g_B = 1.779$, $g_C = 5.5 - 4.0$ , $g_D = 1.973$ and $g_E = 1.945$.)

type frameworks [28], and the signals C and D have been identified there. Signal C is assigned to isolated high-spin Cr(III) ions with $S = 3/2$ in a highly distorted octahedral oxygen ion coordination. This gives rise to large variations in the $E_{ZFS}/D_{ZFS}$ ratios ($0 < E_{ZFS}/D_{ZFS} < 1/3$) leading for $D_{ZFS} > \mu_B B g_C$ to $g$-values between $4.0 - 5.5$. This condition is often met for X-band EPR spectroscopy but may not hold any longer at higher fields [28]. Therefore only a single line at $g = 1.97 - 1.98$ is observed in the Q- and W-band spectra in Figures 4d and 4f. The presence of isolated Cr(III) species is supported by the most likely paramagnetic response in the low-temperature magnetization date of MIL- 100($Al_{0.8}Cr_{0.2}$) (Figure 2b). According to previous assignments, signal D is caused either by AFM exchanged coupled Cr(III)$_2$ pairs [27] or Cr(III)$_x$O$_y$ clusters [28]. Therefore, it seems to be justified to assign signals C and D to Al(III)$_2$Cr(III)$_1$ and Al(III)$_1$Cr(III)$_2$ units in MIL- 100($Al_{0.8}Cr_{0.2}$). Similar mixed metal ion trimers have been identified in MOF MIL-100($Al_{3-x}Fe_x$) [29].

# 4  Conclusion

Magnetization and EPR measurements confirm the formation of AFM coupled $Cr(III)_3$ trimers in the MOF MIL-101(Cr), having an exchange coupling constant $J_0$ = -11.4 $cm^{-1}$. The trimers in the supertetrahedral building units of the MIL-101(Cr) framework are weakly coupled, as indicated by the determined small inter-trimer exchange interaction. The doublet total spin ground state of the $Cr(III)_3$ trimers reveals D-M interaction leading to a characteristic shift of its *g*-value shift at low temperatures. Although the magnetically diluted MOF MIL- $100(Al_{0.8}Cr_{0.2})$ displays AFM properties likewise the temperature-independent *g*-values of the Cr(III) EPR signals indicate that $Cr(III)_3$ trimers are not formed, and the Cr(III) ions are dispersed over the framework. We suggest that Cr(III) is incorporated as paramagnetic $Al(III)_2Cr(III)_1$ and AFM coupled $Al(III)_1Cr(III)_2$ trimeric metal ion units into the MIL-100 framework though we cannot completely exclude the formation of minor isolated and clustered extra-framework chromium species.

# Acknowledgements

**Author contributions**   K.T carried out the X- and Q-band EPR studies.  W-band measurements were performed by A.F and K.T. SQUID measurements were done by M. Z. Team TUD offered MOFs with preliminary characterizations. K.T and A.P wrote the manuscript, and all authors have contributed and approved the final manuscript.

**Funding information:**   The authors, K. Thangavel., D. M. Murphy, and A. Pöppl. from Leipzig and Cardiff universities, gratefully acknowledge funding from the European Union's Horizon 2020 research and innovation programme under the Marie Skłodowska Curie Actions- PARA-CAT(paramagnetic species in catalysis research) grant agreement No 813209. Visit PARACAT to know the technical information about the project and PARACAT-CORDIS for the official information.

# A Supplementary Magnetization results

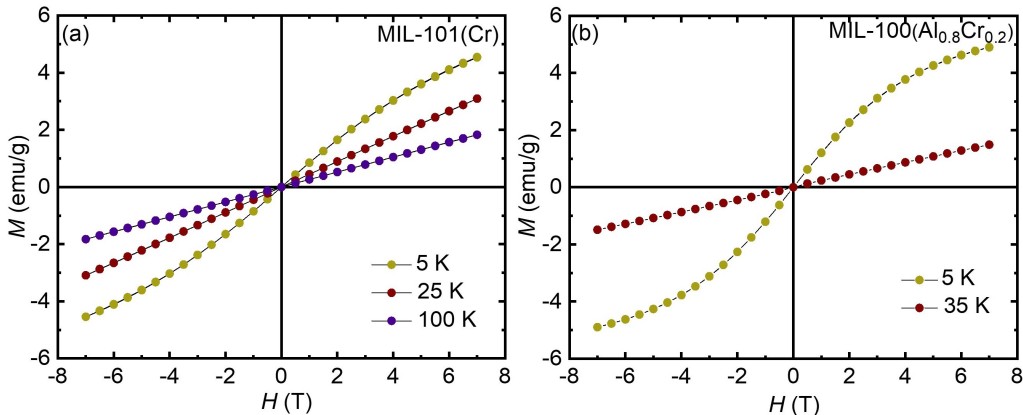

Figure 5: $M$-$H$ curves recorded at 5 K, 25 K and 100 K for (a) MIL-101(Cr) and, 5 K and 35 K for (b) MIL-100($Al_{0.8}Cr_{0.2}$).

The field-dependent magnetization measured at $T = 5$ K, 25 K and 100 K for MIL-101(Cr) and, at $T = 5$ K and 35 K for MIL-100($Al_{0.8}Cr_{0.2}$) show a linear increase of magnetization with the field up to 7 T . No magnetic hysteresis behavior was observed.

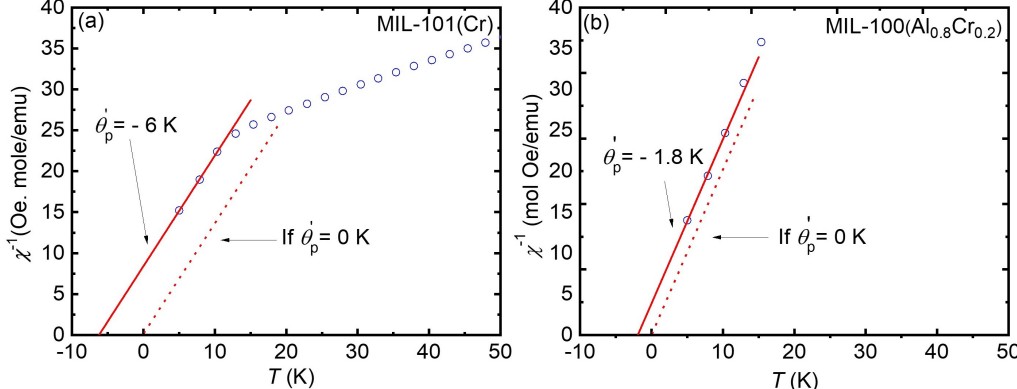

Figure 6: C-W fit on the temperature-dependent reciprocal of the magnetic susceptibility of (a) MIL-101(Cr) and (b) MIL-100($Al_{0.8}Cr_{0.2}$) below $T < 15$ K.

Table 1: Paramagnetic Curie temperature ($\theta_p$), effective magnetic moment ($\mu_{eff}$) and Curie constant values of MIL-100(Cr) and MIL-101($Al_{0.8}Cr_{0.2}$) from SQUID and EPR measurements.

| MOFs | Measurement | $\theta_p$ (K) | $\mu_{eff}$ ($\mu_B$/f.u.) | $C$ |
|---|---|---|---|---|
| **MIL-101(Cr)** | SQUID | -82 | 5.43 | 3.69 |
| | EPR | -70 | - | - |
| **MIL-100** | SQUID | -87 | 3.14 | 1.23 |
| **($Al_{0.8}Cr_{0.2}$)** | EPR | -85 | - | - |

Table 2: Comparison of magnetization values for the MIL-101(Cr) and MIL-100($Al_{0.8}Cr_{0.2}$) MOFs found from $M$-$T$ and $M$-$H$ curves at the 5 K, 25 K, 35 K and 100 K temperatures.

| MOFs | $H$ (T) | $M$ (emu/g) | | | | |
|---|---|---|---|---|---|---|
| | | at 5 K | at 25 K | at 35 K | at 100 K | 300 K |
| **MIL-101(Cr)** | ($M$-$H$)7 T | 4.54 | 3.12 | - | 1.81 | - |
| | ($M$-$H$)0.5 T | 0.44 | 0.23 | - | 0.12 | - |
| | ($M$-$T$)0.5 T | 0.44 | 0.23 | 0.21 | 0.13 | 0.06 |
| **MIL-100** | ($M$-$H$)7 T | 4.95 | - | 1.47 | - | - |
| **($Al_{0.8}Cr_{0.2}$)** | ($M$-$H$)0.5 T | 0.59 | - | 0.12 | - | - |
| | ($M$-$T$) 0.5 T | 0.59 | 0.16 | 0.12 | 0.06 | 0.00 |

# B  Supplementary EPR results

Figure 7 illustrates a series of temperature-dependent X-band EPR data of MIL-100(Cr) and MIL-100($Al_{0.8}Cr_{0.2}$) materials.

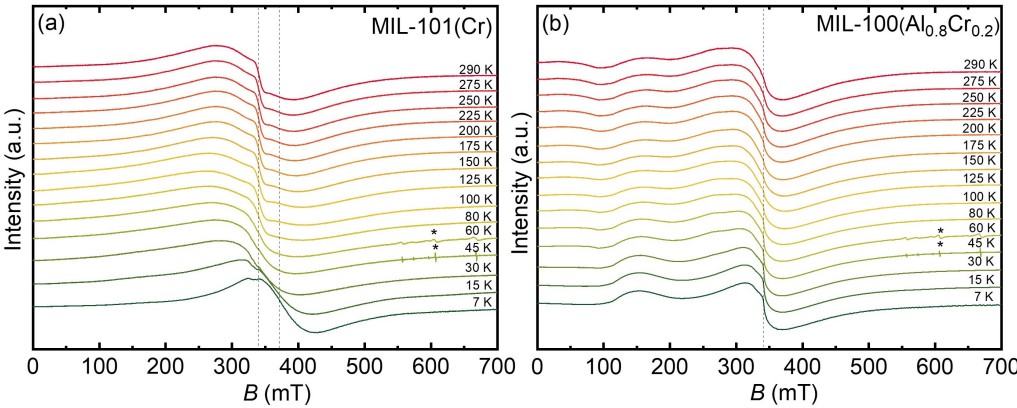

Figure 7: Temperature-dependent X -band EPR data of (a) MIL-101(Cr) and (b) MIL-100($Al_{0.8}Cr_{0.2}$) at temperature ranging from $T = 7$ K to $T = 290$ K(*-corresponds to the gaseous oxygen in the cryostat).

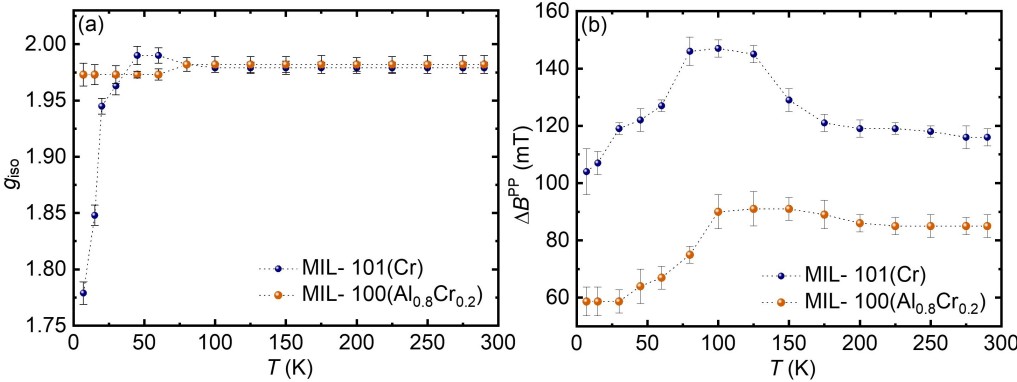

Figure 8: Temperature-dependent (a) $g_{iso}$ trend, and (b) peak to peak width ($\Delta B_{pp}$) of MIL-101(Cr) and MIL-100($Al_{0.8}Cr_{0.2}$) extracted from the X-band temperature-dependent spectra.

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
