# Peer review of "EPR and SQUID interrogations of Cr(III) trimer complexes in the MIL-101(Cr) and bimetallic MIL-100(Al/Cr) MOFs"

_SciPost Physics Proceedings, doi:SciPost Phys. Proc. 11, 016 (2023)_

## Round 1 · Referee Report · Anonymous (Referee 1) · 2023-2-20

Report

The authors present EPR and SQUID measurements on two metal-organic frameworks and elucidate their magnetic structures.

The manuscript is clearly written and presents new material, which is timely and relevant to the community. I therefore think publication in SciPost Conference Proceedings is appropriate. However, there are several issues I suggest the authors address before publication.

- Figure 4 (in the appendix) would be useful to have in Section 2, to clarify which spin interactions are referred to in Eq. 1. Also, the link marked "J1" in the figure is described in Eq1 has having interaction "J0+J1".
- In figure 4 (in the appendix), the upper tetrahedron in panel a clearly shows trimers of octahera surrounding each vertex. I suppose these contain the three spins appearing in panel b (it would be good to indicate this explicitly). Since the arrangement looks symmetric around the vertex, what is the origin of J1 (or J0+J1) being different from J0?
- In the same figure, the lower tetrahedron in panel a has hexagons of octahedra surrounding each vertex, rather than triangles. Why is a trimer model still appropriate for that case? Or equivalently, why do the authors need experimental evidence to rule out the existence of trimers in that case?
- In Eqs. 1, 2, 3 and in the text below them, several spin operators and coefficients should be vectors.
- Below Eq. 1, it is mentioned that the spin interactions are antiferromagnetic. The notation "-J0" would suggest J0>0, in which case the interactions in Eq 1 are actually ferromagnetic?
- Below Eq 1: If the authors write that the degeneracy of the JT=1/2 state is lifted for nonzero J1, I assume everything in the preceding sentences applies only to the case J1=0? If so, this should be clearly stated.
- Below Eq 6, what is the assumption J0>>J1 based on?
- At the end of Sec 3.1, the authors write Fig2 panel b shows an "unspecific response" in the inverse susceptibility, while panel a shows a typical response for trimers. It would be helpful if the authors could point out the qualitative feature that distinguishes these curves?

---

## Round 2 · Author Response

Many thanks to the reviewer for the very helpful comments and suggestions, and the proceedings could certainly be improved by that. Herewith, we attached our answers.

---

## Round 2 · List of Changes

Many thanks to the reviewer for the very helpful comments and suggestions, and the proceedings could certainly be improved by that. Our answers are given below.
1. Figure 4 (in the appendix) would be useful to have in Section 2, to clarify which spin interactions are referred to in Eq. 1. Also, the link marked "J1" in the figure is described in Eq1 has having interaction "J0+J1".
Answer: Yes, the figure has been replaced with section 2 (Figure 1). And the ‘J0+J1’ has been corrected in the figure.
2. In figure 4 (in the appendix), the upper tetrahedron in panel a clearly shows trimers of octahera surrounding each vertex. I suppose these contain the three spins appearing in panel b (it would be good to indicate this explicitly). Since the arrangement looks symmetric around the vertex, what is the origin of J1 (or J0+J1) being different from J0?
Answer: Yes, the figure has been updated with three spins appearing (now, it is Figure 1). The J1 notation changed to J0+J1. We assume that the Cr(III) ions in the trimer unit are equally separated. J1 is valid only when there is a difference in distance for one of the Cr(III) ions. However, we consider J1 to be negligible.
3. In the same figure, the lower tetrahedron in panel a has hexagons of octahedra surrounding each vertex, rather than triangles. Why is a trimer model still appropriate for that case? Or equivalently, why do the authors need experimental evidence to rule out the existence of trimers in that case?
Answer: There is no hexagons of octahedra. It is again trimers of octahedra for both MIL-100 and MIL-101 MOFs. Probably, the previous figure was misleading. The better figure has been updated now (Figure 1).
The reason for the two trimer systems: Firstly, we wanted to study the magnetically diluted MIL-100(Al/Cr):20% Cr for the catalytic application. But the EPR signatures of MIL-100(Al/Cr) weren’t sufficient to conclude the trimer/dimer preference of Cr(III) in the MIL-100 framework. Therefore, we picked pure MIL-101(Cr) in comparison with MIL-100, and the EPR results confirm that there is no trimer formation in MIL-100(Al/Cr) based on our observation of low-temperature DM interaction in the Cr(III)-trimer systems of MIL-101(Cr) .
4. In Eqs. 1, 2, 3 and in the text below them, several spin operators and coefficients should be vectors.
Answer: It is updated.
5. Below Eq. 1, it is mentioned that the spin interactions are antiferromagnetic. The notation "-J0" would suggest J0>0, in which case the interactions in Eq 1 are actually ferromagnetic?
Answer: The notation of exchange interaction is always debatable, and different authors follow different conventions. Based on the convention from the EPR books, the positive spin Hamiltonian of exchange interaction is meant for antiferromagnetic coupling. When J < 0, the spin Hamiltonian becomes positive, which indicates antiferromagnetic interaction.
6. Below Eq 1: If the authors write that the degeneracy of the JT=1/2 state is lifted for nonzero J1, I assume everything in the preceding sentences applies only to the case J1=0? If so, this should be clearly stated.
Answer: We rewrote the sentence as below.
“Eqn.1 results in five degenerate energy
levels corresponding to the total spin states ST = 1/2, 3/2, 5/2, 7/2 and 9/2 of the trimer with
the twofold degenerate ST = 1/2 state is the ground state in the case of antiferromagnetically
(AFM) coupled trimers when J1 = 0. The degeneracy of the ST = 1/2 state will be further
lifted only when J1 ̸= 0. However, J1 is considered to be negligible since Cr(III)
ions in the trimer unit is assumed to be in equal distance.”
7. Below Eq 6, what is the assumption J0>>J1 based on?
Answer: Because we assume that Cr(III) ions are almost equally separated and the J1 term is negligible.
8. At the end of Sec 3.1, the authors write Fig2 panel b shows an "unspecific response" in the inverse susceptibility, while panel a shows a typical response for trimers. It would be helpful if the authors could point out the qualitative feature that distinguishes these curves?
Answer: Now, it is figure 2 Figure 3; according to Honda et al. ( J. Phys.
Soc. Jpn. 61(10), 3773 (1992) ), the typical sharp steep drop of Cr(III) trimer is not observed in the Cr(III) diluted MIL-100(Al0.8Cr0.2) MOF. We mentioned that as an unspecific response in MIL-100 which does not indicate Cr(III) trimer formation. However, we improved that sentence.
1. Figure 4 (in the appendix) would be useful to have in Section 2, to clarify which spin interactions are referred to in Eq. 1. Also, the link marked "J1" in the figure is described in Eq1 has having interaction "J0+J1".
Answer: Yes, the figure has been replaced with section 2 (Figure 1). And the ‘J0+J1’ has been corrected in the figure.
2. In figure 4 (in the appendix), the upper tetrahedron in panel a clearly shows trimers of octahera surrounding each vertex. I suppose these contain the three spins appearing in panel b (it would be good to indicate this explicitly). Since the arrangement looks symmetric around the vertex, what is the origin of J1 (or J0+J1) being different from J0?
Answer: Yes, the figure has been updated with three spins appearing (now, it is Figure 1). The J1 notation changed to J0+J1. We assume that the Cr(III) ions in the trimer unit are equally separated. J1 is valid only when there is a difference in distance for one of the Cr(III) ions. However, we consider J1 to be negligible.
3. In the same figure, the lower tetrahedron in panel a has hexagons of octahedra surrounding each vertex, rather than triangles. Why is a trimer model still appropriate for that case? Or equivalently, why do the authors need experimental evidence to rule out the existence of trimers in that case?
Answer: There is no hexagons of octahedra. It is again trimers of octahedra for both MIL-100 and MIL-101 MOFs. Probably, the previous figure was misleading. The better figure has been updated now (Figure 1).
The reason for the two trimer systems: Firstly, we wanted to study the magnetically diluted MIL-100(Al/Cr):20% Cr for the catalytic application. But the EPR signatures of MIL-100(Al/Cr) weren’t sufficient to conclude the trimer/dimer preference of Cr(III) in the MIL-100 framework. Therefore, we picked pure MIL-101(Cr) in comparison with MIL-100, and the EPR results confirm that there is no trimer formation in MIL-100(Al/Cr) based on our observation of low-temperature DM interaction in the Cr(III)-trimer systems of MIL-101(Cr) .
4. In Eqs. 1, 2, 3 and in the text below them, several spin operators and coefficients should be vectors.
Answer: It is updated.
5. Below Eq. 1, it is mentioned that the spin interactions are antiferromagnetic. The notation "-J0" would suggest J0>0, in which case the interactions in Eq 1 are actually ferromagnetic?
Answer: The notation of exchange interaction is always debatable, and different authors follow different conventions. Based on the convention from the EPR books, the positive spin Hamiltonian of exchange interaction is meant for antiferromagnetic coupling. When J < 0, the spin Hamiltonian becomes positive, which indicates antiferromagnetic interaction.
6. Below Eq 1: If the authors write that the degeneracy of the JT=1/2 state is lifted for nonzero J1, I assume everything in the preceding sentences applies only to the case J1=0? If so, this should be clearly stated.
Answer: We rewrote the sentence as below.
“Eqn.1 results in five degenerate energy
levels corresponding to the total spin states ST = 1/2, 3/2, 5/2, 7/2 and 9/2 of the trimer with
the twofold degenerate ST = 1/2 state is the ground state in the case of antiferromagnetically
(AFM) coupled trimers when J1 = 0. The degeneracy of the ST = 1/2 state will be further
lifted only when J1 ̸= 0. However, J1 is considered to be negligible since Cr(III)
ions in the trimer unit is assumed to be in equal distance.”
7. Below Eq 6, what is the assumption J0>>J1 based on?
Answer: Because we assume that Cr(III) ions are almost equally separated and the J1 term is negligible.
8. At the end of Sec 3.1, the authors write Fig2 panel b shows an "unspecific response" in the inverse susceptibility, while panel a shows a typical response for trimers. It would be helpful if the authors could point out the qualitative feature that distinguishes these curves?
Answer: Now, it is figure 2 Figure 3; according to Honda et al. ( J. Phys.
Soc. Jpn. 61(10), 3773 (1992) ), the typical sharp steep drop of Cr(III) trimer is not observed in the Cr(III) diluted MIL-100(Al0.8Cr0.2) MOF. We mentioned that as an unspecific response in MIL-100 which does not indicate Cr(III) trimer formation. However, we improved that sentence.

---

## Editorial Decision

published